# LARM: Large Auto-Regressive Model for Long-Horizon Embodied Intelligence

## Abstract

Due to the need of interacting with the world, embodied agents are required to possess comprehensive task-relevant knowledge, long-horizon planning capability, and a swift response speed. Large language models (LLMs), owing to their rich general knowledge, recently achieve promising results in open-world embodied tasks, like the world exploration in Minecraft. However, the outputs of LLMs are descriptive sentences or code, which are slow to generate and not end-to-end, as a translator is required to translate the LLM outputs into actions to perform. To address these limitations, we introduce the large auto-regressive model (LARM). LARM leverages environment observations as input and predicts subsequent actions in an auto-regressive manner. Compared with LLM based methods, LARM directly predicts the next skill for execution according to the current observation. In addition, considering that the commonly adopted training paradigms do not reflect the mutual influence and dependency between actions and observations, we develop a novel data format named auto-regressive node transmission structure and assemble a corresponding dataset to train LARM. Combining these techniques, LARM successfully harvests enchanted equipment in Minecraft, which demands significantly more complex decision-making chains than the highest achievements of prior best methods. Besides, the speed of LARM is $6.8\times$ faster than LLMs with similar parameter volume.

## 1 Introduction

In recent years, remarkable progress has been achieved in various artificial intelligence (AI) fields LeCun et al. (2015) like computer vision He et al. (2016) and natural language processing Kenton & Toutanova (2019), but most of them lack the capacity to physically interact with the real world. To address this disconnect, the concept of embodied AI is introduced Chrisley (2003). Early embodied agents are predominantly developed on simulation platforms for specific tasks such as object grasping and indoor navigation Savva et al. (2019). While notable advancements are achieved, these agents tend to be specialist models confined to isolated tasks Huang et al. (2023). To overcome this limitation, recent studies, including this work, employ Minecraft Baker et al. (2022); Fan et al. (2022); Guss et al. (2019) as a benchmark to explore embodied agents with open-ended objectives and long-horizon reasoning chains.

The early methods for developing such agents primarily rely on reinforcement learning, the exploration of which is inefficient and results in limited performance Yuan et al. (2023). Recent works begin to investigate the use of large language models (LLMs) Brown et al. (2020). Owing to the extensive general knowledge and formidable reasoning capabilities of LLMs, these methods demonstrate promising results with significantly reduced domain-specific engineering efforts Wang et al. (2023a). Nevertheless, LLMs continue to exhibit a number of limitations, rendering them ill-suited for embodied AI tasks. First of all, the outputs of LLMs are usually sentences or code Zhao et al. (2023), which are generated through iterative token prediction. This prediction mode necessitates $N$ inference operations for $N$ tokens, resulting in sluggish response speed, which is critical for embodied applications. Secondly, recent research suggests that a huge model size is important for an LLM to master the ability to generate correct descriptions or code Achiam et al. (2023). Nevertheless, the computing resource for deploying embodied agents is usually very limited. Moreover, LLMs are commonly trained through web data based text completion, which does not reflect how the actions affect the environment and how the environment update alters the optimal action.

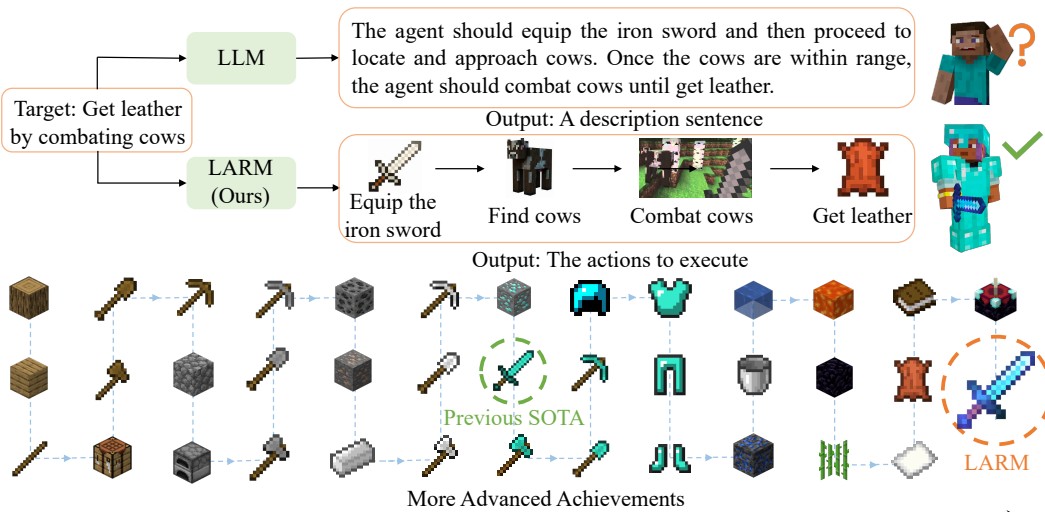

Figure 1: LARM is the first method that achieves enchanted diamond equipment in Minecraft. Different from LLMs, LARM directly predicts subsequent skills in an end-to-end manner.

To address the limitations of LLM, we propose **L**arge **A**uto-**R**egressive **M**odel (LARM). Taking both text information and vision observation as input, LARM engages in real-time interaction with the environment and predicts subsequent actions in an auto-regressive manner. Instead of generating a descriptive sentence composed of multiple tokens, LARM directly produces a single token to determine the next action. In addition, LARM shares a similar basic network architecture with existing LLMs, enabling us to utilize LLM weights to initialize the parameters of LARM. In this way, LARM is equipped with both rich general knowledge and high inference efficiency.

For training, we first employ a 34G multi-modal dataset Fan et al. (2022), sourced from Wiki Minecraft webpages, to provide LARM with general knowledge about the Minecraft world. However, this web dataset does not sufficiently describe the mutual dependency relations among the environment, observations, and actions. To bridge this gap, we introduce a unique data organization structure termed as **A**uto-**R**egressive **N**ode **T**ransmission **S**tructure (ARNTS). Within ARNTS, each data sample is conceptualized as a node, encompassing text information, multi-view images, and the expected subsequent action. A chronological dependency exists among these nodes. The process of accomplishing a complex target can be interpreted as transmissions between different nodes. Adhering to this data structure, we manually collect an ARNTS dataset comprising 2,589 data samples, which cover diversified targets, weathers, and biomes. Leveraging this dataset to train the LARM agent, the agent can effectively learn the dependency knowledge among various information types.

Through integrating the aforementioned efforts, we develop an agent capable of crafting enchanted diamond tools. As shown in Fig. 1, crafting enchanted diamond tools requires a far more intricate step chain compared to existing state-of-the-art (SOTA) methods Wang et al. (2023a); Zhao et al. (2023), which are only capable of crafting standard diamond tools. Besides, the inference speed of LARM is $6.8\times$ faster on average compared to an LLM with comparable parameter volume. This work reveals a new path of developing data-driven embodied intelligence.

## 2 RELATED WORK

**Embodied benchmarks.** Embodied AI recently garners significant attention due to its potential to bridge the gap between virtual algorithms and physical interactions Bredeche et al. (2018). The works in embodied AI primarily focus on enhancing the interaction of agents with the environment, and many benchmarks are established Duan et al. (2022). Existing benchmarks can be categorized into two classes, simulator-based and game-based. Simulator-based benchmarks are established upon popular simulation platforms, such as MuJoCo Todorov et al. (2012) and Isaac Gym Makoviychuk et al. (2021). Through complex engineering efforts on dynamics modeling, these simulation platforms mimic the interaction kinematics between agents and manipulated objects in the real world well. Some recent benchmarks designed based on these simulation platforms include

RLbench James et al. (2020), HumanoidBench Meser et al. (2024), etc. Nevertheless, running these simulation platforms could take much computing resources even though only simulating a very limited range of the environment. Due to this problem, existing works conducted on these benchmarks usually only focus on low-level motion control Zhuang et al. (2023). By contrast, we mainly study the high-level and long-horizon embodied tasks in this work and assume the low-level policies have been well developed. Another line of embodied AI benchmarks is the game-based ones. Different from simulator-based benchmarks, game-based benchmarks do not pay much attention to simulating motion dynamics Anderson et al. (2018). Alternatively, they mainly focus on providing environments for performing high-level actions Savva et al. (2019). Hence, this category of benchmarks is more suitable for exploring how to develop agents with complex decision-making capabilities.

**Minecraft agents.** Minecraft is an open-ended platform suitable for exploring building agents with long-horizon planning capabilities Fan et al. (2022). It simulates diverse weathers, biomes, and mobs in an unlimited 3D virtual world and provides well-designed low-level APIs of basic actions PrismarineJS (2013). According to the output formats of models, we classify existing works into two categories, *i.e.*, atom-based and skill-based. Early methods mostly adopt the atom-based paradigm, which means the output of the model is directly a low-level atom action, *e.g.*, a short movement, a mouse click, or a keyboard press Frazier & Riedl (2019). The advantage of this paradigm is its control flexibility. However, due to the huge potential decision space, such atom-based agents are quite challenging for optimization, and thus these works pay their main attention to devising strategies for alleviating the optimization complexity Scheller et al. (2020).

Instead of using atom actions, the output of methods based on the skill-based paradigm is directly the skill, such as chopping down a tree or crafting a table. The skill could be a well-trained policy based on reinforcement learning or provided APIs. As the available skills help agents avoid the need to learn low-level skills, the training difficulty is greatly alleviated. Therefore, the works based on this paradigm concentrate better on how to boost the long-term scheduling ability of agents. Voyager Wang et al. (2023a) is representative among the skill-based methods. It successfully crafts diamond tools using a training-free system based on GPT-4 Achiam et al. (2023). Similarly, DEPS Wang et al. (2023b) develops a more comprehensive LLM system, yielding satisfactory results. However, these methods rely heavily on existing knowledge of LLMs. If an LLM does not have accurate knowledge about this task, the agents cannot complete it successfully. Besides, LLMs solely support text input, which is inadequate for encapsulating environmental information. To surmount this hurdle, certain studies train large vision-language models through fine-tuning LLMs Zhao et al. (2023); Feng et al. (2024). Despite these advancements, the output of these models remains descriptive text. In this work, we aim to explore how to develop agents with long-term scheduling intelligence.

**Large language models.** LLMs draw broad attention from the research and industrial communities due to their rich general knowledge and the ability to generate the answers to diverse kinds of questions Chang et al. (2024). Early LLMs mainly serve as pre-training weights for downstream tasks Devlin et al. (2019). GPT-3 emerges as a milestone in the evolution of LLMs, as it takes the next token prediction problem as the pre-training task and showcases remarkable open-world generalization capabilities across diverse tasks Brown et al. (2020). Subsequently, the fine-tuning of GPT-3 using reinforcement learning with human feedback leads to the creation of ChatGPT, a model that displays an impressive breadth of general knowledge OpenAI (2023). Further advancements result in the creation of GPT-4, reinforcing the advantages of increasing model size and training data Achiam et al. (2023). Although these LLMs present remarkable performance, their model weights are mostly inaccessible to the research community. To bridge this gap, some notable open-source works are released. These works train models with fewer parameters while still achieving promising results, such as LLaMA Touvron et al. (2023). However, a significant limitation of LLMs is their inability to interpret information in images, which are vital for humans to perceive the world.

To overcome this problem, researchers devise strategies that inject vision information into LLMs and enable LLMs to perceive images. The common method is fine-tuning a small amount of network parameters using numerous language-image data pairs to bridge the representation gap between text and images Ding et al. (2023). In this way, some large vision-language models like LLaVA Liu et al. (2024) and Flamingo Alayrac et al. (2022) are derived. While these models can describe the content in provided images, they lack the capability to interact with the real world, which needs the models to predict the correct actions to execute according to the environment status. To address this limitation, some works like EmbodiedGPT boost the performance of LLMs on scheduling actions through fine-tuning models using chain of thought data Mu et al. (2024). The fine-tuning process is

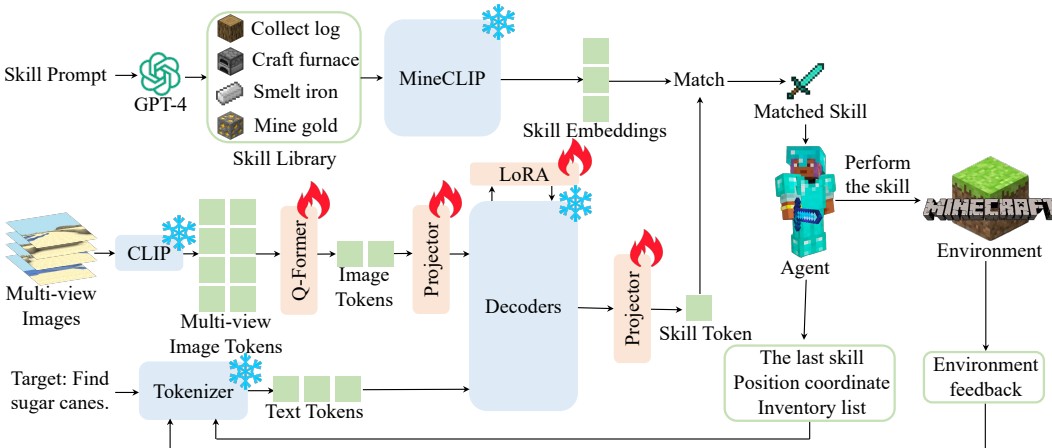

Figure 2: The overall framework of LARM. In this framework, the network takes the target task description, multi-view images, agent information, and environment feedback as input to predict a skill token. The skill token is matched with the skill embeddings, which are generated based on a pre-prepared skill library, to select the optimal skill. Then, the agent performs this skill, which helps the agent one step closer to completing the target task and changes the environment.

often based on a technique named LoRA Hu et al. (2021), which greatly alleviates the optimization challenge and computational burden compared with fine-tuning the whole model. Although these embodied LLMs have achieved promising results to some extent, their output is still in the format of descriptive sentences, which is not naturally consistent with the action space.

## 3 METHOD

### 3.1 PROBLEM FORMULATION

What we study in this work can be conceptualized as an auto-regressive prediction problem involving long sequences, and is effectively framed as a Markov Decision Process, symbolized by $\mathcal{E} = (\mathcal{S}, \mathcal{A}, \mathcal{P})$. In this formulation, $\mathcal{S}$ is the space encompassing all possible states of the environment, $\mathcal{A}$ is the set of feasible actions that the policy can take, and $\mathcal{P}$ represents the probability distribution that governs state transitions given a current state and action. At any discrete time step $t$, the environment resides in a state $s_t \in \mathcal{S}$, and the corresponding observation $o_t$ by a policy $\pi$ is a function of this state, expressed as $o_t = f(s_t)$. This observation $o_t$ is then utilized to select the subsequent action according to $a_t \sim \pi(o_t)$, where $a_t \in \mathcal{A}$. After the action $a_t$ is executed, the environment transitions to the next state following the probability distribution $s_{t+1} \sim \mathcal{P}(s_{t+1} \mid s_t, a_t)$.

In tackling the long-horizon prediction problem, the objective is to navigate through a sequence of intermediate states $s_1, s_2, \ldots, s_{T-1}$ to ultimately reach the target state $s_T$ at the final time step $T$. This requires the policy to generate a series of actions $a_0, a_1, \ldots, a_{T-1}$ such that each action $a_t$ transitions the environment from state $s_t$ to the next state $s_{t+1}$, adhering to the dynamics prescribed by the transition probability distribution $\mathcal{P}$. Formally, the trajectory through the state sequence is constructed iteratively. Starting from an initial state $s_0$, the policy selects an action $a_0$ based on the initial observation $o_0 = f(s_0)$ leading to a transition $s_1 \sim \mathcal{P}(s_1 \mid s_0, a_0)$. This process is repeated iteratively such that at any time step $t$, the action $a_t$ selected from the policy $\pi$ depends on the observation $o_t = f(s_t)$, thereby driving the state evolution $s_{t+1} \sim \mathcal{P}(s_{t+1} \mid s_t, a_t)$. It is crucial that each intermediate state $s_t$ is accurately predicted and achieved in sequence to ensure the policy attains the target state $s_T$ by the final time step $T$.

### 3.2 LARGE AUTO-REGRESSIVE MODEL

In this work, we model the policy $\pi$ by the proposed large auto-regressive model, which contains billions of parameters and predicts subsequent skills to execute in an end-to-end manner. Its framework is illustrated in Fig. 2. The skill library is generated by GPT-4, and the description of every

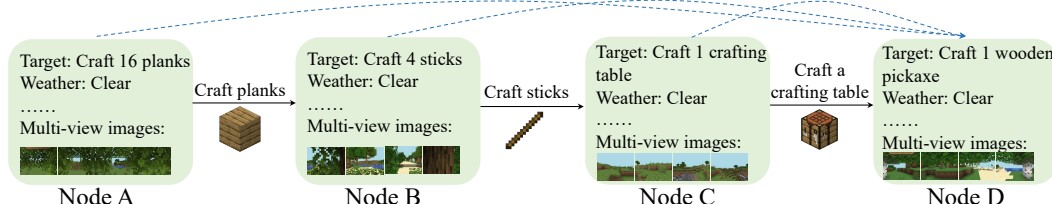

Figure 3: For ARTNS, the procedures of performing a multi-step task are interpreted as a graph. In different nodes, the agent perceives the environment information and selects a skill to execute. During an edge linking two nodes, the agent performs the selected skill and its status transfers from the start node to the end node. Besides, there exist dependencies among various nodes. For example, as marked by the blue dotted lines, Node D depends on the first three nodes, because crafting 1 wooden axe requires the planks, sticks, and crafting table in Minecraft.

skill is mapped to an embedding vector with the length of $L$ based on MineCLIP Fan et al. (2022). LARM selects a skill through predicting a vector closest to the corresponding skill embedding.

The real-time observation $o_t$ of LARM at the timestamp $t$ consists of both text and image information. The text information includes the target task description, agent information, and environment feedback. Specifically, the agent information contains the last skill performed by the agent, the current 3D position coordinate of the agent in the Minecraft world, and the inventory list (a list of the resources that the agent owns). For the environment feedback, two kinds of information are included, *i.e.*, whether the last skill performed by the agent is executed successfully and the game response message to that skill. All the text information is tokenized into text tokens by a frozen tokenizer, the weight of which is initialized from BERT Kenton & Toutanova (2019).

Different from previous Minecraft agents that take a monocular image as input Zhao et al. (2023), LARM adopts $N$ views of images in different directions, providing a more comprehensive description of the surrounding environment. The $N$ multi-view images are tokenized as $N \times N^I$ image tokens by a frozen CLIP encoder. However, since there are $N$ images, the number of obtained tokens is increased by $N$ times and would result in a significant computational burden if using all these tokens as input to the subsequent decoders. To resolve this problem, we utilize a trainable Q-Former Li et al. (2023) module to reduce the number of image tokens from $N \times N^I$ to $N^I$, as shown in Fig. 2. Besides reducing image tokens, this Q-Former module is also helpful for fusing the feature existing in multi-view images. After the Q-Former, the image tokens are further transformed by a trainable projector linking the vision branch with decoders. The decoders take both text and image tokens as input to conduct feature interaction. Given that the decoders contain numerous parameters and are challenging to train, we initialize the parameters from the LLaVA weight Liu et al. (2023) and freeze them during training. Additionally, a trainable LoRA Hu et al. (2021) is applied to assist the model in understanding Minecraft knowledge.

The output of the decoders is multiple tokens and we compress them into a single token with a trainable projector. We name the token produced by this projector as skill token, which contains the information about which skill should be executed according to the observation. By conducting cosine distance Nguyen & Bai (2010) based matching between the skill token and the many pre-generated skill embeddings, the most similar skill is selected for the agent to perform. The chosen skill alters both the status of the agent and the environment. LARM repeats the above process in an auto-regressive way until an *End* token is matched, marking the end of this skill execution chain.

## 3.3 TRAINING PROTOCOL

We develop a two-stage protocol to train LARM, including Wiki pre-training and ARTNS fine-tuning. Although the parameters of LARM are initialized from LLaVA, its initial knowledge about the Minecraft world is very limited. Therefore, we first employ 34G webpage data Fan et al. (2022) crawled from Wiki to pre-train LARM. Specifically, we remove the text comprising fewer than 30 letters in the webpage data. In this way, 63,666 sentences are obtained. The sentences are matched with the closest images on webpages and provided to the model. In Wiki pre-training, we mask a part of these sentences and train the model to predict the masked words. This helps the pre-trained

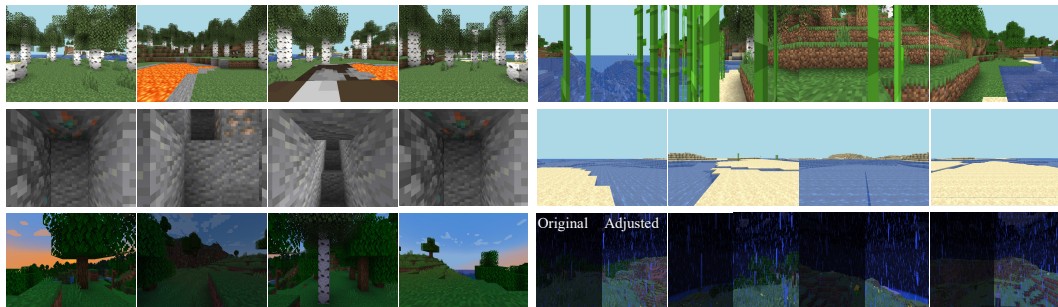

Figure 4: Some examples of the collected multi-view images, which cover diverse tasks, biomes, weather, etc. The bottom right group of images is a rainy night case so its images are dim. We adjust its color contrast for clearer visualization, and the input to LARM is still in the original form.

model gain a better understanding of Minecraft. The model in Wiki pre-training shares the same network structure and training process as LLaVA. Subsequently, in ARTNS fine-tuning, we employ the decoder weight of the pre-trained model to initialize the parameters of the LARM decoders shown in Fig. 2. The experimental results suggest that the devised Wiki pre-training effectively improves the convergence speed of LARM.

During ARTNS fine-tuning, we fine-tune the LARM policy with our defined ARNTS data. In ARTNS, we represent the process of performing a multi-step task as a sequential chain, which includes nodes and edges that link these nodes. As illustrated in Fig. 3, the nodes denote the agent and environment status $\{s_t\}_{t=0}^T$, and edges represent the corresponding selected skills $\{a_t\}_{t=0}^T$. At timestamp $t$, when a skill $a_t$ is performed, the status transitions from a node $s_t$ to the next node $s_{t+1}$. Notably, for $i < j$, reaching $s_j$ does not necessarily require first arriving in $s_i$. There could be multiple viable transition paths, reflecting the huge decision-making challenges in Minecraft. For example, as shown in Fig. 3, making a crafting table requires planks but not sticks, which means Node C depends on the completion of the target in Node A but not Node B. In addition, a node could depend on multiple previous nodes, like arriving in Node D relies on first reaching Node A, Node B, and Node C, simultaneously. In accordance with the ARTNS format, we manually operate an agent to complete various tasks in Minecraft and record the data. Through collecting data in diverse biomes and weather, we obtain a dataset consisting of 2,589 data pairs. Each pair contains the exploration trajectory ID, data pair ID, time, weather, biome name, target, agent position, agent inventory list, the name of the last performed skill and its corresponding execution result, the next skill, and multi-view images. Some multi-view image examples are visualized in Fig. 4. Then, we annotate the dependency relations among nodes, which are represented as a graph like the blue dotted lines illustrated in Fig. 3. This graph describes the skills in some nodes are prerequisites for other nodes, and the agent should learn this knowledge to understand how to schedule future skills.

## 4 EXPERIMENTS

### 4.1 EXPERIMENTAL SETUP

The LARM hyper-parameters $N$, $H$, $W$, $L$, and $N^I$ are set to 4, 480, 640, 512, and 1,024, respectively. LARM is built with 32 decoders, which contain about 7B parameters. The epoch number and learning rate are set to 1 and 2e-4 for Wiki pre-training, and 6 and 2e-5 in ARTNS fine-tuning. The training batch size is 16. Other training details, like the choice of optimizer and learning scheduler, follow LLaVA. The basic skills are implemented based on Mineflayer.

In Minecraft, due to the lack of a widely accepted benchmark, different methods often employ varying training pipelines, testing protocols and training data. Therefore, to fully reveal the effectiveness of LARM, we compare LARM with methods adopting different testing protocols. In addition, we make sure that the testing protocol adopted by LARM is the same or more challenging than the compared methods. In this way, the experimental results that reveal the superiority of LARM are guaranteed to be convincing. To be more transparent about the differences among various methods, we elaborate on the details of all compared methods, including MineAgent Fan et al. (2022),

Table 1: Performance comparison with reinforcement learning skill based methods.

| Task | MineAgent | Plan4MC | LLaMA-Rider Base | LLaMA-Rider | LARM |
|---|---|---|---|---|---|
| Harvest stick | 0.00 | 0.30 | 0.23 | 0.43 | 1.00 |
| Harvest crafting table | 0.03 | 0.30 | 0.37 | 0.67 | 1.00 |
| Harvest bowl | 0.00 | 0.47 | 0.73 | 0.97 | 1.00 |
| Harvest chest | 0.00 | 0.23 | 0.67 | 0.77 | 1.00 |
| Harvest wooden sword | 0.00 | 0.47 | 0.63 | 0.10 | 1.00 |
| Harvest furnace | 0.00 | 0.37 | 0.00 | 0.17 | 1.00 |
| Harvest stone stairs | 0.00 | 0.47 | 0.00 | 0.57 | 1.00 |
| Harvest stone sword | 0.00 | 0.10 | 0.00 | 0.00 | 1.00 |
| Harvest iron ingot | 0.00 | 0.47 | 0.03 | 0.13 | 1.00 |
| Harvest bucket | 0.00 | 0.20 | 0.00 | 0.00 | 1.00 |
| Harvest iron sword | 0.00 | 0.20 | 0.00 | 0.00 | 1.00 |
| Harvest beef | 0.33 | 0.43 | 0.03 | 0.03 | 1.00 |
| Harvest mutton | 0.35 | 0.33 | 0.00 | 0.03 | 1.00 |
| Harvest diamond sword | 0.00 | 0.00 | 0.00 | 0.00 | 1.00 |
| Harvest enchanted sword | 0.00 | 0.00 | 0.00 | 0.00 | 0.70 |

Table 2: Performance comparison with LLM based methods on the tech tree mastery.

| Achievement | ReAct | Reflexion | AutoGPT | VOYAGER | STEVE | LARM |
|---|---|---|---|---|---|---|
| Wooden Tool | 0/3 | 0/3 | 3/3 | 3/3 | 3/3 | 30/30 |
| Stone Tool | 0/3 | 0/3 | 3/3 | 3/3 | 3/3 | 30/30 |
| Iron Tool | 0/3 | 0/3 | 3/3 | 3/3 | 3/3 | 28/30 |
| Diamond Tool | 0/3 | 0/3 | 0/3 | 1/3 | 3/3 | 27/30 |
| Enchanted Tool | 0/3 | 0/3 | 0/3 | 0/3 | 0/3 | 21/30 |

LLaMA-Rider Feng et al. (2024), Voyager Wang et al. (2023a), ReAct Yao et al. (2022), Reflexion Shinn et al. (2023), AutoGPT autogpt, and STEVE Zhao et al. (2023).

Specifically, MineAgent is the baseline method provided by Minddojo. It first fine-tunes CLIP Radford et al. (2021) based on numerous web data and uses the fine-tuned CLIP to guide the training of reinforcement learning algorithms. Plan4MC is a reinforcement learning based method. It splits a task into basic skills and trains an agent to learn them one by one in a hierarchical way. LLaMA-Rider is an LLM obtained by fine-tuning LLaMA. It first makes the agent explore the environment to collect data. Then, they adopt the collected data to fine-tune LLaMA in a supervised manner. Voyager is a training-free method implemented based on GPT-4. Its main contribution is designing a multi-step prompt generation pipeline. Its skills are implemented based on Mineflayer. When a target task is given, Voyager prompts GPT-4 to know which skill should be executed and gradually realizes the target. ReAct, Reflexion, and AutoGPT are LLMs and can reason about which skill should be performed through multi-step question answering. STEVE is a large vision-language model. In this work, a dataset including both videos and text-image pairs is gathered and utilized to fine-tune LLaMA, and then the fine-tuned model can invoke pre-defined skills.

## 4.2 COMPARISON WITH PREVIOUS METHODS

**Tech tree mastery comparison.** In this experiment, we compare the tech tree mastery of LARM with previous counterparts, and the details of these methods have been explained in Section 4.1. The task completion success rates of LARM and reinforcement learning skill based methods, which include MineAgent, Plan4MC, and LLaMA-Rider, are reported in Table 1. In this table, following previous works Wang et al. (2023a); Feng et al. (2024), except for the tasks "harvest diamond sword" and "harvest enchanted sword" where the compared methods are tested 3 times, all the methods are run 30 times in other tasks. The shown numbers are the success rates of the methods in completing these tasks successfully. The best method is highlighted in light gray. A variety of tasks are tested and the best method is marked in light gray. It can be observed that the reinforcement learning skill based methods display significantly poorer performance compared with LARM, which can complete almost all these tasks with a success rate of 100%. Besides, except LARM, all these methods fail to craft a diamond sword or enchanted sword. This is because these methods are still struggling with how to schedule the low-level actions well.

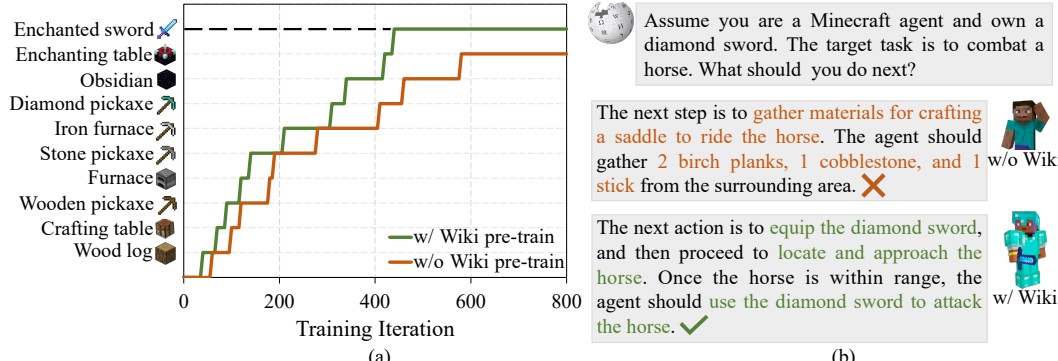

Figure 5: Analysis of the effect of Wiki pre-training. (a) The most advanced items achieved by the models with and without Wiki pre-training after various iterations of ARTNS fine-tuning. It can be observed that the model converges more quickly and presents better performance with ARTNS fine-tuning. (b) The model initialized from LLaVA lacks knowledge about Minecraft and gives a false answer to the question. By contrast, the model after Wiki pre-training answers correctly. The key points of the false and correct responses are highlighted in bronze and green, respectively.

We further compare LARM with LLM based methods in Table 2. In this table, the results of the compared methods are obtained from the work Voyager Wang et al. (2023a). These methods are run for 3 times to compute success rates. To make the result more convincing, we test LARM for 30 times. Notably, all the compared methods demand tens of prompting iterations, while LARM obtains the enchanted tool achievement with only 1 prompt, which reflects the efficiency of LARM. The best method is highlighted in light gray. First of all, it can be found that the results of the methods in Table 2 are significantly better than those in Table 1. This observation indicates that LLM-based agents generally outperform reinforcement learning skill based ones. Secondly, LARM is the only method capable of crafting enchanted tools, which requires a complex skill execution chain. This observation indicates that LARM presents stronger long-horizon scheduling ability compared with existing methods, and this advantage arises from that the output of LARM is more consistent with the requirement of embodied AI tasks.

**Inference efficiency.** Besides the tech tree mastery achievement, inference efficiency is also important. However, many of the aforementioned methods are implemented based on LLM APIs. The inference speed of these methods is significantly affected by network latency, while LARM is a model deployed in local machines. Additionally, the GPUs used by these methods may vary and are uncontrollable. Therefore, directly comparing their inference speeds is meaningless. Thankfully, the methods built upon LLM share a similar Transformer architecture. Thus, it is reasonable to compare LARM with an LLM of the same parameter volume to validate inference efficiency. In line with this thought, we compare the inference time costs of LARM and the LLaVA-1.6 model with both 7B parameters. They are tested in 100 cases to compute the average inference time, with each case corresponding to a node in Fig. 3. The output of LLaVA is prompted to be as concise as possible. This experiment is performed using one RTX3090 GPU. Through this experiment, we find that the average inference time of LLaVA on the 100 cases is 5.78 seconds while LARM is 0.85 seconds, which indicates that LARM is 6.8× more efficient. The high efficiency of LARM stems from its distinctive design, which enables the LLM to generate a single token for every skill.

### 4.3 STUDY ON PERFORMANCE DYNAMICS

In this part, we study the changing dynamics of the highest achievement the agent can obtain and the effect of Wiki pre-training. The results are depicted in Fig. 5. Specifically, in Fig. 5 (a), we compare the most advanced items obtained by the models after different iterations of ARTNS fine-tuning. Achieving an upper item indicates more superior performance, and the two curves correspond to models with and without Wiki pre-training. Comparing these two curves, we can conclude that Wiki pre-training accelerates the convergence of LARM and contributes to better performance. Moreover, it can be found that our method presents high training efficiency. For example, LARM harvests the

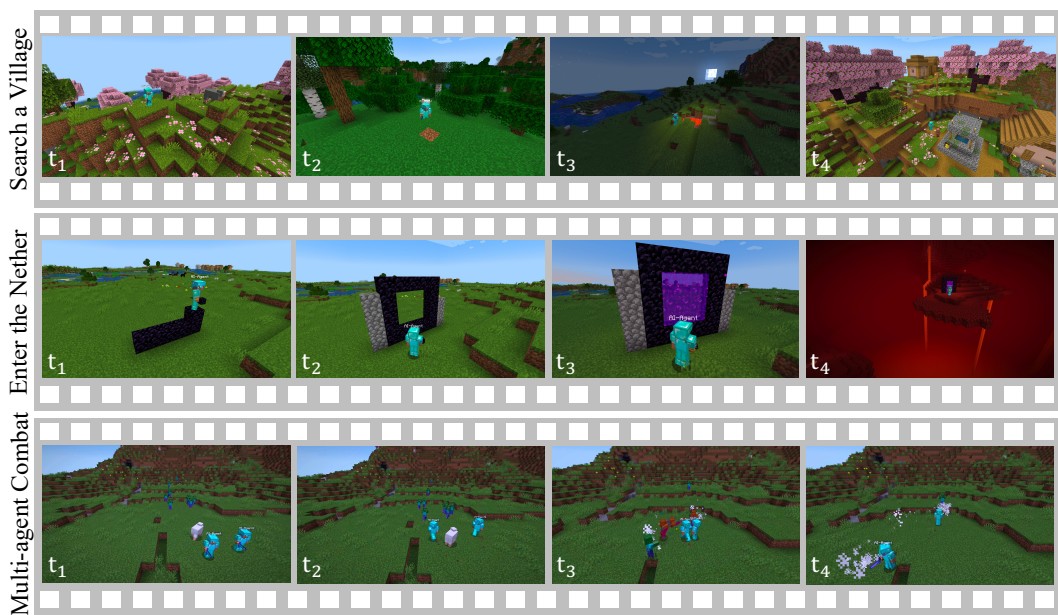

Figure 6: More behavior example illustrations of LARM, which include traveling a long distance to find a village, building a nether portal, and then entering the nether, multiple agents collaborate with each other to combat zombies.

iron tool achievement after fewer than 300 training iterations. By contrast, previous reinforcement learning based methods demand thousands of exploration iterations to learn to craft a wooden tool.

Fig. 5 (b) presents the responses of the models with and without Wiki pre-training to a question about Minecraft. The model without Wiki pre-training is exactly the original LLaVA-1.6-7B model. The key points of these two answers are marked in different colors. We can find that although the question is about how to combat a horse when a diamond sword is in the inventory, the model without Wiki pre-training provides an answer about riding a horse instead. Besides, the provided recipe for crafting a saddle is incorrect. By contrast, the model after Wiki pre-training accurately and in detail describes the steps for combating a horse. This demonstrates that Wiki pre-training effectively enriches the general knowledge of LARM about Minecraft.

### 4.4 CASE STUDY

In Fig. 6, we present additional examples of LARM learned behaviors. In the first case, the agent travels through various biomes to find a village, which demonstrates that LARM can continuously explore the open world until finding the desired object. For the second one, the agent builds a nether portal and enters the Nether through it. This case suggests the promising construction capability of LARM. In the last example, two agents collaborate with each other to combat a large group of zombies, confirming that multiple LARM models can efficiently cooperate together.

## 5 CONCLUSION

In this work, we have focused on strategies of how to design and implement long-horizon embodied intelligence in an efficient manner. We have introduced the concept of the large auto-regressive model and implemented a corresponding model LARM, an embodied agent predicting subsequent skills in an auto-regressive manner. To train LARM, a new data organization structure has been proposed and a corresponding dataset has been collected. Extensive experiments have been conducted to validate the effectiveness of LARM. Compared with previous methods, LARM has presented stronger long-horizon scheduling ability and faster inference speed. LARM is the first method capable of crafting enchanted diamond tools in Minecraft. We hope this work to significantly contribute to the development of high-level embodied intelligence.

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

## A  VIDEO DEMO

We present a video in the Supplementary Material to show how the agent successfully harvests enchanted diamond tools. The video showing diverse basic capabilities is named as "LARM-enchanted-submission.mp4".

