# OpenReview forum: "LARM: Large Auto-Regressive Model for Long-Horizon Embodied Intelligence"
_ICLR.cc/2025/Conference — ICLR 2025 Conference Withdrawn Submission_

### Official Review · Reviewer_cxPa · 2024-10-22

**Soundness:** 3
**Presentation:** 3
**Contribution:** 2
**Rating:** 5
**Confidence:** 3

**Summary:**

This paper introduces a novel auto-regressive method for long-horizon planning called LARM, which is an extension of LLaVA with a trainable LoRA module. This auto-regressive approach directly predicts subsequent actions rather than generating long-context task descriptions, enabling end-to-end skill generation. To address the limitations of environmental dependency in web data, the paper also defines a graph-like data node structure (ARNTS). This structure allows the agent to learn how to schedule future skills by following a sequential chain in multi-step tasks. The paper demonstrates remarkable improvements compared to recent state-of-the-art methods in the Minecraft environment. LARM achieves 100% success on 14 out of 15 skills, significantly outperforming baselines such as MineAgent and LLaMA-Rider. Notably, LARM is the first method to successfully craft an enchanted tool, which requires a complex chain of steps to achieve.

**Strengths:**

1. The writing in this paper is well-structured, and the experimental results demonstrate the effectiveness of the proposed method.
2. LARM shows impressive performance in both skill prediction and tech tree mastery, indicating promising improvements in long-term decision-making chains.
3. LARM significantly enhances the efficiency of LLMs-based methods deployed on local machines, showcasing the advantages of its end-to-end subsequent skill prediction approach.

**Weaknesses:**

1. The authors assert that learning the dependency relationships among nodes is crucial for the agent's understanding of the long-horizon chain of skill prediction. Could the authors provide detailed information on the implementation of this node dependency learning process, particularly regarding the blue dotted lines in Figure 3?

2. In this paper, LARM utilizes multi-view images as input for visual tokens, and I agree that this approach can enhance the visual perception of the surrounding environment, particularly for embodied agents. However, it would be beneficial to conduct an ablation analysis on the number of views and compare the results with those from monocular images to further refine the training design.

3. The skill library in LARM is largely based on the one built for Voyager. In Voyager's paper, they conducted a zero-shot generalization evaluation on unseen tasks, demonstrating the potential of a lifelong skill library to expand skills in an open-world environment. Since LARM's skill library is pre-prepared, I wonder if LARM can similarly adapt to new tasks in a zero-shot manner as the skill library expands?

4. This paper discusses inference efficiency in the comparison section, but the comparison setup is unclear. From my understanding, LARM is built on LLaVA, with additional trainable modules like Q-Former, LoRA, and a projector. In this context, the differences in inference efficiency compared to vanilla LLaVA are primarily due to these added modules and the length of the text context. As a result, the comparison lacks clarity and is not convincing, making it difficult to draw firm conclusions about LARM's inference efficiency.

5. This paper has demonstrated that Wiki pre-trained LLaVA can learn general knowledge in the Minecraft world, which raises an interesting question about the performance of Wiki-pretrained LLaVA in predicting the next skills. While the authors assert that LLM outputs are typically limited to sentences or code, I do not agree that the output format limit the use of LLMs. For example, a prompt design strategy could be employed, using question prompt templates for LLaVA to predict fixed-length skill keywords. These keywords could then be projected into MineCLIP embeddings as query tokens for the skill library. Therefore, I recommend that the authors conduct additional analyses of the Wiki pre-trained LLaVA to thoroughly explore the capabilities of VLMs.

**Questions:**

Questions:
1.This paper introduces a new data structure called Auto-Regressive Node Transmission Structure (ARNTS). However, it also mentions a similar term, ARTNS, which appears to be a typographical error for ARNTS and can lead to confusion while reading. If ARTNS refers to a technique in another paper, please provide the appropriate citation.

While the authors provide a detailed comparative analysis with baseline methods, the paper still lacks point-by-point examinations of their proposed settings to demonstrate the overall pipeline. This raises some concerns, which I have outlined in the above questions.
I will reevaluate the paper's rating after carefully considering the perspectives of other reviewers and the authors' responses. I am also willing to increase my score if the authors address my concerns.

---

### Official Review · Reviewer_7jmP · 2024-11-03

**Soundness:** 2
**Presentation:** 3
**Contribution:** 2
**Rating:** 3
**Confidence:** 5

**Summary:**

To address the problem that LLM-based embodied agent policy requires post-processing, the authors proposed LARM: large auto-regressive model to predict the next skill for execution, taking in environment feedback and observations.  They also proposed auto-regressive node transmission structure to reflect the correlation between actions and observation. They conducted experiments on Minecraft and completed some complex decision-making tasks with faster inference speed.

**Strengths:**

-The authors proposed LARM to solve complicated decision-making tasks in Minecraft and achieved good results than previous models.

-The authors pinpointed the potential problem of lacking mutual dependency between actions and observation

-The inference speed of this method is faster than those LLM-based ones.

**Weaknesses:**

-When considering decision-making agents, the authors missed important benchmarks in related work.

-There is no evidence showing that propose method can generalize to other domains.

-The proposed method requires engineering adaption to new domains.

-The proposed model consists of many components, yet the individual performance contribution and their necessity are not shown in any
ablation studies at all.

-The proposed model is not novel enough, a combination of foundation models.

**Questions:**

-How does the agent perform the skill in Minecraft? Is it hard-coded and executed deterministically?

-Does this method generalize beyond Minecraft in other decision-making environments (e.g. ALFRED)?

-Having a finite skill space might limit the generalizability of the proposed framework. What is the authors response to the statement?

---

> ### Public Comment · ~Zhuofan_Wen4 · 2024-11-18
>
> Although this paper indeed utilizes numerous foundation models, I believe that pre-encoding skills and then leveraging rapid matching through learning is a common and effective approach in embodied intelligence control. If we have a similar skill library in text form that enables quick action execution via CLIP matching, this approach holds some merit and thus should not be considered "not novel enough." Additionally, the original text already explains how the skill library is generated: "The skill library is generated by GPT-4, and the description of each skill is mapped to an embedding vector with the length of L based on MineCLIP." Furthermore, the reasoning behind using Minecraft as the platform for agent research has already been addressed in works such as Malmö and MineDojo.

---

### Official Review · Reviewer_jMv8 · 2024-11-04

**Soundness:** 2
**Presentation:** 2
**Contribution:** 3
**Rating:** 5
**Confidence:** 3

**Summary:**

Recently, embodied agents have been considered one of the promising and attractive research directions that are able to communicate with their surroundings and do complex open-world tasks. However, these agents may require comprehensive task knowledge, long-term planning, and rapid response capabilities to interact effectively with the world. Although LLMs excel in open-world tasks leveraging their rich knowledge, their outputs are slow and not directly executable, requiring translation into actions. To overcome these limitations, this paper introduces the large auto-regressive model (LARM), which directly predicts the next action based on environment observations in an auto-regressive manner. Unlike LLMs, LARM forecasts actionable skills immediately, reducing latency. The authors also introduce an auto-regressive node transmission data format to train LARM, emphasizing dependencies between actions and observations.

**Strengths:**

- The proposed method can be fully embedded into the device without the need for a call from a proprietary LLM API.
- Inference speed seems to be faster than the LLava-1.6-7B. (But the comparison looks a little bit tricky.)
- Reasonable architectural design and pre-training/fine-tuning processes.
- Solid performance over the Minecraft simulation benchmark.

**Weaknesses:**

- The grounds for using Q-former architecture for visual token compression are not sufficiently validated. If this Q-former structure aims for token compression into a fixed length, there should be a number of competitive techniques, including token sparsification, merging, pooling, convolution layers, MLPs, Mixture-of-experts, etc. I expect analyses that provide reasonable observations and insights into why this employed design is more appropriate than those nonparametric and parametric potential counterparts for Long-Horizon Embodied Intelligence.

- Also, the effect of other components, such as training the LoRA or projector weights, is not clearly verified. At the same time, the discussion of the 'single skill token' is also not very clear. It is also possible to  1) produce multiple skill tokens, 2) perform the Hungarian algorithm to match 'multiple' skill tokens with relevant skill embeddings, and then 3) select the best-matched skill based on some importance/uncertainty metrics. More discussions and insights/evidence on the use of single-skill tokens would be helpful for the design choice of the proposed approach.

- Efficiency comparison sounds not that reliable to me. Even baselines with LLM API calls have a fluctuation in inference time depending on network latency; it would be meaningful to compare the wall-clock running time/cost among baselines and the proposed methods under stable network conditions.

- detailed explanations of the usage of GPT-4 for the skill library and the examples/format of /environment feedback' and 'the last skill position coordinate inventory list' are not provided. I believe the design choice of this structure also affects the model performance; the authors should provide detailed information in the appendix.

**Questions:**

Please see the weakness.

---

### Official Review · Reviewer_JNC3 · 2024-11-05

**Soundness:** 3
**Presentation:** 3
**Contribution:** 2
**Rating:** 5
**Confidence:** 3

**Summary:**

This work introduces the Large Auto-Regressive Model (LARM) for long-horizon tasks, focusing on the Minecraft environment. LARM is built upon various existing techniques, including GPT-4, LoRA, and Q-Former. Given a specific target task, LARM takes multi-view images along with the corresponding task text as input to predict the next action. The predicted skill token is then used to match an existing skill set generated by GPT-4, allowing for selecting the optimal skill. The authors claim this approach is efficient as it predicts the next action directly. Additionally, to train LARM, the work introduces a data format known as ARNTS to manage sequential chains. Experimental results indicate that LARM outperforms existing reinforcement learning-based methods.

**Strengths:**

- The work is well-organized and includes numerous details that help readers understand the LARM process.
- Besides LARM, the work defines a dataset for training the model, enabling LARM to learn sequential actions effectively.
- A demonstration of LARM is also provided.

**Weaknesses:**

- The method relies heavily on existing techniques, which undermines the novelty of the work.
- The tasks in the experiments are not particularly challenging, raising concerns about the generalizability of LARM.
- The formulation of LARM is relatively simple and closely resembles imitation learning. Aside from reinforcement learning methods, are there any imitation learning-based baselines to compare with LARM?
- Furthermore, have the baselines in Tables 1 and 2 been fine-tuned with the same data as LARM?

**Questions:**

Please see weaknesses.

---

### Note · Authors · 2024-11-18

I have read and agree with the venue's withdrawal policy on behalf of myself and my co-authors.